# Interface Structure and Band Alignment of CZTS/CdS Heterojunction: An Experimental and First-Principles DFT Investigation

**DOI:** 10.3390/ma12244040

**Published:** 2019-12-05

**Authors:** Sachin Rondiya, Yogesh Jadhav, Mamta Nasane, Sandesh Jadkar, Nelson Y. Dzade

**Affiliations:** 1The School of Chemistry, Cardiff University, Cardiff, Wales CF10 3AT, UK; RondiyaS@cardiff.ac.uk; 2National Center for Nanoscience and Nanotechnology, Mumbai University, Mumbai 400098, India; nano4yash@gmail.com; 3Department of Physics, Savitribai Phule Pune University, Pune 411007, India; nasanemamta@gmail.com (M.N.); sandeshjadkar@gmail.com (S.J.)

**Keywords:** earth-abundant materials, kesterite-CZTS, H_2_S annealing, band alignment and offsets, density functional theory, solar cell

## Abstract

We report a phase-pure kesterite Cu_2_ZnSnS_4_ (CZTS) thin films, synthesized using radio frequency (RF) sputtering followed by low-temperature H_2_S annealing and confirmed by XRD, Raman spectroscopy and XPS measurements. Subsequently, the band offsets at the interface of the CZTS/CdS heterojunction were systematically investigated by combining experiments and first-principles density functional theory (DFT) calculations, which provide atomic-level insights into the nature of atomic ordering and stability of the CZTS/CdS interface. A staggered type II band alignment between the valence and conduction bands at the CZTS/CdS interface was determined from Cyclic Voltammetry (CV) measurements and the DFT calculations. The conduction and valence band offsets were estimated at 0.10 and 1.21 eV, respectively, from CV measurements and 0.28 and 1.15 from DFT prediction. Based on the small conduction band offset and the predicted higher positions of the VB_max_ and CB_min_ for CZTS than CdS, it is suggested photogenerated charge carriers will be efficient separated across the interface, where electrons will flow from CZTS to the CdS and and vice versa for photo-generated valence holes. Our results help to explain the separation of photo-excited charge carriers across the CZTS/CdS interface and it should open new avenues for developing more efficient CZTS-based solar cells.

## 1. Introduction

The development of scalable, sustainable and economical solar cells demands stable, earth-abundant, non-toxic, and highly efficient absorber materials. Kesterite-copper-zinc-tin chalcogenides Cu_2_ZnSnS_4_ (CZTS) and Cu_2_ZnSnSe_4_ (CZTSe) have recently emerged as promising absorber materials for scalable production of low-cost and environmentally friendly thin-film photovoltaics (PV). CZTS is ideally positioned as a next-generation PV material because it combines an optimal bandgap of 1.5 eV, and a high optical absorption coefficient of 10^4^ cm^−1^ in the visible light region, with the predicted theoretical power conversion efficiency (PCE) more than 30% [1]. Moreover, the bandgap of CZTS material can be easily engineered through alloying with dopant elements such as Cd, Se, Ge, Mg, Ni, Co or Ag [2]. Most importantly, the quaternary compound semiconductors (CZTS and CZTSe) are composed of naturally abundant and non-toxic elements, making them ideal candidates for the large-scale production of eco-friendly thin-film solar cells compared to the current range of thin-film solar cell absorbers including CdTe and Cu(In,Ga)Se_2_ (CIGS) [3,4]. Like other recently developed multi-metallic photocatalysts [5,6], the unique physiochemical and electronic properties of CZTS materials make them also attractive heterogeneous catalyst for the electrocatalytic hydrogen evolution reaction (HER) and photocatalytic degradation of polluting dyes [7,8]. Owing to efficient charge separation, heterostructures of copper-zinc oxide (CZO) have shown better activity towards hydrogen evolution reaction under sunlight than individual CuO and ZnO materials [5].

Notwithstanding their superior optical and electronic properties, to date, the highest reported power conversion efficiencies (PCEs) of Cu_2_ZnSnSe_4_ (CZTS), Cu_2_ZnSnSe_4_ (CZTSe) and Cu_2_ZnSn(S,Se)_4_ (CZTSSe) have peaked at 8.5% [9], 11.6% [10], and 12.6% [9,11], respectively, which are rather substantially lower than the reported efficiencies Cu(In,Ga)Se_2_ (21.7%) and CdTe (21.0%) devices [9]. The stagnation of the efficiency of CZTS devices has been attributed mainly to a large V_OC_ deficits [12,13,14], which have been suggested to result from several factors including but not limited to high-defect states at the CZTS/buffer interface [15], and co-existence of secondary phases [16] that cause high recombination of photogenerated carriers [17,18]. Unfavourable band alignment at the CZTS/buffer interface, which limits efficient charge separation is another commonly reported cause for the limiting the V_OC_ and efficiency of CZTS devices [19].

To minimize the V_OC_ loss and further improve the PCE of CZTS solar cells, it is important to gain fundamental atomic-level knowledge about the structure of CZTS/CdS heterojunction interface and accurately predict the magnitude of the conduction band offset (CBO) and valence band offset (VBO) at the interface. The band alignment is arguably one of the most important properties of the semiconductor–semiconductor heterostructure as it dictates the barrier height, and, therefore, controls charge carrier generation and transport phenomena at interfaces and characteristics of the devices employing these interfaces. Due to their small extent in one dimension and typical location buried within bulk materials, interfaces are difficult to resolve or access by purely experimental means, but a synergistic experimental–computational approach promises to offer an in-depth understanding structure–property relationship of the CZTS/CdS heterojunction interface. Although the structure, defect chemistry, and charge carrier dynamic properties of CZTS have been investigated in previous studies [20], there exist limited investigations dedicated to providing fundamental understanding of the microscopic structure and composition of the CZTS/CdS interface. Besides, the exact positions of the band edges at the CZTS/CdS interface, which dictates the functionality of fabricated devices is poorly understood.

In this paper, we studied the interface structure and band offsets at the CZTS/CdS interface using a combination of experimental methods and first-principles calculations. A staggered type II band alignment with conduction (0.1 eV) and valence (1.21 eV) band offsets at the CZTS/CdS interface is determined using Cyclic Voltammetry (CV) measurements and is corroborated by our first-principles density functional theory (DFT) calculations. The small conduction band offset (CBO) of points to efficient charge separation in the CZTS/CdS interface which is vital for efficient device operation. The atomic-level insights derived in the present study should be relevant to other related scientific disciplines such as photocatalysis.

## 2. Experimental and DFT Calculations Details

### 2.1. Experimental Details

The CZTS thin films were deposited on corning #7059 substrates using indigenously developed RF magnetron sputtering system. A CZTS target of 4-inch diameter (99.99%, VINKAROLA, Norcross, Georgia, USA), 3 mm thick was used for the deposition and was kept facing the substrate holder at ~7 cm away. In order to get film uniformity, the substrates were kept rotating during the sputtering process using a stepper motor with variable speed controller. The deposition pressure and substrate temperature were kept constant at 2 × 10^−2^ Torr and 400 °C, respectively. The CZTS thin films were annealed in H_2_S atmosphere (flow rate of 20 sccm) at 250 °C for 30 min, to improve their grain size. Next, we prepared CdS thin film as a buffer layer and using the radio-frequency magnetron sputtering technique, a CdS thin film of ∼300 nm thickness was deposited on the corning substrate. A 4-inch diameter target (99.99% pure, Vin Karola Instrument, Norcross, Georgia, USA) was employed for the deposition of CdS thin films. The distance between the substrate holder and target was ~7 cm. The base pressure of the vacuum chamber was evacuated to be 1 × 10^−3^ torr, and then, the high-purity Ar gas (99.99%) with a flow of 20 sccm was used as the work gas. The deposition time was optimized for 10 min.

The crystal structures and structural properties of the CZTS thin films were studied by X-ray Diffractometer (Bruker D8 Advance, Karlsruhe, Baden-Wurttemberg, Germany), Raman spectroscopy (Renishaw Raman microscope, Wotton-under-Edge, Gloucestershire, UK) and XPS measurements were performed using an XPS instrument (ESCALAB 250, Thermo Fisher Scientific, Waltham, Massachusetts, USA) with an Al Ka (*hυ* = 1486.6 eV) radiation and all the binding energies were calibrated by the C 1s peak (284.6 eV). The surface morphology properties were investigated using FEI-Nova Nano SEM 450 field-emission scanning electron microscopy (FE-SEM, FEI, Hillsboro, Oregon, USA), whereas the surface topology properties were determined using conductive atomic force microscopy (C-AFM) (Multimode 8.0, Bruker, Santa Barbara, USA)). The electrochemical measurements (cyclic voltammetry (CV)) were performed using Metrohm potentiostat/galvanostat, Autolab PGSTAT 100 (Utrecht, The Netherlands). The CV characterization was performed using a homemade cell setup in which a commercial Au disk electrode (CHI Instruments, Austin, Texas, USA; 2 mm diameter), an Ag wire, and a Pt wire loop were used as working, quasi-reference, and counter electrodes, respectively. The electrodes were cleaned using dilute nitric acid. For controlled measurements, a blank CV was recorded in the tetrabutyl ammonium perchlorate–acetonitrile (TBAP-ACN) mixture, and then on the working electrode loaded with samples to the cell. The working electrode was prepared by drop-casting 200 µL of dispersion in ACN with net concentration of 1.0 mg/mL. The scan rate for the CV measurements is 100 Mv/s. At the end of each experiment, the potentials were recalibrated with respect to the normal hydrogen electrode (NHE), using ferrocene as an internal standard [21,22].

### 2.2. Computational Details

To provide an atomic-level insight into the structure and composition of the CZTS/CdS interface and to better understand the energy band alignment derived from the CV characterizations, we carried out electronic structure DFT calculations as implemented in the Vienna ab initio simulation package (VASP) [23,24]. The PBE functional [25] was used for geometry optimizations while for the electronic structures and optical calculations, the screened hybrid functional HSE06 with 25% Hartree–Fock exchange was used [26]. The projected augmented wave (PAW) method was used to describe the interactions between the valence and core electrons [27,28]. The CZTS/CdS heterojunction interfaces were constructed with (2 × 2)-CZTS (001) and a (3 × 2)-CdS (100) supercells, which ensured that the lattice mismatch at the interface is less than 5%. The Brillouin Zone of bulk CZTS was sampled using the Γ-centred 5 × 5 × 3 k-mesh with energy cut-off of 600 eV, whereas the constructed interface structures were adapted 5 × 5 × 1. Geometry optimizations have performed based on conjugate-gradient algorithm until the residual Hellmann–Feynman forces on all relaxed atoms reached 10^−3^ eV Å^−1^. The density of states (DOS) has calculated using tetrahedron method [29] with Bloch correction and the optical properties through the frequency-dependent dielectric matrix.

## 3. Results and Discussion

X-ray diffraction pattern of the as-deposited and H_2_S-annealed CZTS thin films (Figure 1a) show four distinct peaks at 2θ~28.5°, 30.1°, 31.5° and 47.5°, which corresponds to the (112), (103), (200), and (220) planes, respectively. These peaks are in excellent agreement with the standard JCPDS data # 26-0575 and previously reported XRD data for kesterite-CZTS [30]. The higher intensity of the (112) peak relative to the other peaks shows strong preferential growth in the (112) orientation plane. The H_2_S-annealed CZTS film exhibit higher and sharper intensity of the (112) peak, demonstrating improvement in the crystallinity of the film. The average crystallite size of the CZTS films increased from 7.7 nm to 9.0 nm after H_2_S annealing as estimated from the Debye–Scherrer equation [31]. Raman spectra of the as-deposited and H_2_S-annealed CZTS films (Figure 1b) shows only one major peak at ~335–336 cm^−1^, in good agreement with previous reported Raman data for kesterite CZTS films [32]. No other peaks that may correspond to ZnS (350 cm^−1^), Cu_2−x_S (473 cm^−1^) and Cu_2_SnS_3_ (303 cm^−1^) were observed in the Raman spectra, suggesting the formation of phase pure kesterite CZTS. The valence states of Cu, Zn, Sn, and S elements in the as-synthesized kesterite CZTS films (Figure 1c), which identifies the presence of Cu 2*p*, Zn 2*p*, Sn 3*d* and S 2*p* were analysed through X-ray photoelectron spectrometry (XPS) measurements. The spectrum of Cu 2p shows two peaks at 931.6 (2*p*_3/2_) and 951.4 (2*p*_1/2_) with a splitting of 19.8 eV, which is consistent with the standard separation (19.9 eV) of Cu(I) [33,34]. The Zn 2*p* spectra is dominated by 2p_3/2_ and 2p_1/2_ states located at 1021.3 and 1044.6 eV, respectively, with a separation of 23.3 eV, which can be assigned to Zn(II) [35]. The peaks of Sn 3*d* show binding energies at 486.12 (2p5/2) and 494.61 eV (2p3/2) with a split orbit of 8.5 eV, which is in good agreement with the value of Sn(IV) [35]. The S 2p peaks are located at 160.86 (2p3/2) and 161.87 eV (2p1/2), which are consistent with the binding energy of S in the sulfide state in CZTS [36].

The FE-SEM images of the as-deposited and H_2_S-annealed CZTS films (Figure 2a) reveal their surface morphology and cross-section. As shown in Figure 2a(i), the as-deposited CZTS films are smooth and highly dense without defects such as cracks, pinholes, and protrusion with small grain sizes. The H_2_S-annealed CZTS film (Figure 2a(ii)) show agglomeration of the small grains to form larger grains with slightly increased surface roughness. The thickness of the as-deposited CZTS film measured from the cross-sectional FE-SEM image (Figure 2a(iii)) was estimated at ~798 nm. The C-AFM and 3D AFM topographic images of the of as-deposited and H_2_S-annealed CZTS films (Figure 2b(i)) indicate that the as-deposited CZTS films exhibit grain size of ~20 nm with surface roughness ~2.29 nm. Under an applied bias voltage of 0.2 V, all the grains become electrically conducting with the bias distributed over the entire sample (Figure 2b(ii)). After H_2_S annealing the topographic image of CZTS clearly shows an increase in the grain size, estimated at ~100 nm (Figure 2b(iv)) with a surface roughness of ~3.17 nm (Figure 2b(vi)). The C-AFM image of the H_2_S-annealed film (Figure 2b(v)) shows that the maximum current occurs along the grain boundaries (GBs), an indication of reduced charge carrier recombination of at the GBs. It can thus be conjectured that larger GBs of the H_2_S-annealed CZTS films acts as channels for current to flow rather act as recombination sites.

As seen from Figure 3a, the Raman spectra for the CdS thin film has two dominant peaks, first centered ~300 cm^−1^ and second ~600 cm^−1^ corresponding to 1 LO (longitudinal optical) and 2 LO phonon peaks, respectively. The atomic force microscopy (AFM) image (Figure 3b) gives surface features and topology information. The AFM image shows the CdS thin film to have smooth, homogeneous and uniform surface with uniformly oriented spherical nanoparticles.

The valence and conduction band edges of the H_2_S-annealed CZTS films and cadmium sulfide (CdS), which is commonly employed as heterojunction partner for CZTS in PV device fabrication were determined through cyclic voltammetry measurements. The CV measurements permit the estimation of the electrochemical bandgap and the electron affinity (EA) of CZTS and CdS films. Figure 3c shows the CV data for CZTS, with two peaks at −3.62 V (marked as A_1_) and at −5.01 V (marked as C_1_) over repeated cycles, which correspond to the anodic and cathodic peaks, respectively. The values obtained for the CZTS band edges are in very good agreement with the earlier report by Jadhav et al. [21].

For CdS (Figure 3d), the anodic and cathodic peaks are observed at −3.72 V (marked as A_1_) and at −6.22 (marked as C_1_). The potential difference between the anodic and cathodic peaks of CZTS (1.39 V) and CdS (2.5 V) is the measure of their bandgaps. To determine the band discontinuity at the CZTS/CdS interface, the exact positions of the conduction and valence bands with respect to the vacuum level are calculated using the relations:
E_VBM_ = −IP = −(E_peak-oxidation_ + E_ref_)(1)
E_CBM_ = −EA = −(E_peak-reduction_ + E_ref_)(2)
where IP is the ionization potential [37]. Ferrocene was used as the internal reference E_ref_ (4.5 eV vs NHE), and E_peak-reduction_ and E_peak-reduction_ energies corresponds to A1 and C1 peaks, respectively. According to Equations 1 and 2, the EA and IP of the CZTS are obtained at 3.62 and 5.01, respectively, whereas for CdS, they are estimated at 3.72 and 6.22, respectively.

Consequently, a type-II band alignment is predicted for the CdS/CZTS heterojunction as shown in Figure 4a. The offset at the conduction band (CBO) and valence band (VBO) of the CdS/CZTS heterojunction are estimated at 0.1 eV and 1.21 eV, respectively, with the conduction band minimum (CBM) of CZTS found to be higher than that of CdS. This suggests that photogenerated electrons will flow from CZTS to CdS and vice versa for photogenerated valence band photogenerated holes.

To gain further insights into the nature (structure and composition) of the CZTS/CdS interface and understand the staggered type-II band alignment at the CZTS/CdS heterojunction from the CV measurements, we have simulated CZTS/CdS heterostructure and derived atomic-level insights into atomic ordering at the interface, its stability and band offset. CZTS crystallizes in both the kesterite (ks) and stannite (st) crystal structures but the ks-CZTS is commonly reported as the ground-state structure. The energy difference per atom between ks-CZTS and st-CZTS is, however, only of few meV [38,39], indicating that both phases can coexist in experimental samples. We have, therefore, modelled both phases of CZTS in the tetragonal structure; kesterite in the $I\bar{4}$ space group (Figure 5a) and stannite in the $I\bar{4}$2*m* space group Appendix A. First, we calculated the electronic structures (band structure and partial density of states (PDOS)) of the isolated bulk crystals using the screened hybrid HSE06 functional as shown in Figure 5b for ks-CZTS and Appendix A for st-CZTS. An analysis of the electronic band structures of ks-CZTS and st-CZTS revealed that their conduction band minimum (CBM) and valence band maximum (VBM) is located at the same high-symmetry Γ-points in the Brillouin zone, making both CZTS phases direct bandgap semiconductors. The band gap of the ks-CZTS and st-CZTS is predicted at 1.49 eV (Figure 5b,c) and 1.46 eV (Appendix A)), respectively, in good agreement with the experimental data and previous theoretical predictions [30,40,41,42]. It can be seen from the predicted PDOS that the VBM consists mainly of Cu-3*d* and S-3*p* atomic orbitals, whereas the CBM primarily comprises of Sn-5*s* and S-3*p* orbitals. The band gap of the CdS (modelled in the hexagonal wurtzite structure) is calculated to be 2.45 eV (the crystal structure, band structure, and density of states are provided in Appendix A).

The CZTS/CdS heterojunction interfaces were constructed with a (2 × 2)-CZTS(001) and (3 × 2)-CdS(100) supercells, which ensured that the lattice mismatch at the interface is less than 5%. The optimized structure of the ks-CZTS/CdS and st-CZTS/CdS heterostructures are shown in Figure 5d and Appendix A reveal that the interfaces are composed mainly of strong covalent Cu–S and Cd–S interactions. The Cu–S and Cd–S interatomic bond distances at the ks-CZTS/CdS interface are 2.229 Å and 2.578 Å, respectively, compared to 2.304 Å and 2.597 Å at the st-CZTS/CdS interface. The thermodynamic stability of the heterojunctions was evaluated through interfacial adhesion energy, calculated as *E*_ad_ = (*E*_CdS/CZTS_ − (*E*_CdS_ + *E*_CZTS_)/*S*, where *E*_CdS/CZTS_, *E*_CdS_ and *E*_CZTS_ are the total energy of the CZTS/CdS heterostructure with interface surface area *S*, the individual ground state relaxed total energy of the CdS and CZTS surfaces, respectively. The adhesion energy of the ks-CZTS/CdS and st-CZTS/CdS interfaces was calculated to be −0.189 eVÅ^−^^2^ and −0.201 eVÅ^−^^2^, respectively, which indicates both interfaces are thermodynamically stable. Visualization of the differential electron density isosurface reveals electron density accumulation in the interface regions, which is consistent with the newly formed Cu–S and Cd–S chemical bond at the interface.

The magnitude of band offsets (BOs) at the heterojunction interfaces were calculated by the potential-line-up approach based on the relation: $BO=\Delta E_{v,c}+\Delta V,$ where $\Delta E_{v}$ is the difference in the VBM of the two materials and ΔV results from the line-up of the macroscopic average of the self-consistent electrostatic potential across the interface. The electrostatic potential $V\left(\bar{r}\right)$ was derived from solving the solving the Poisson equation, whereas the planar-averaged potential $\bar{V}\left(z\right)$ was determined from the equation: $\bar{V}\left(z\right)=\frac{1}{S}V\left(\bar{r}\right)dxdy$, where S is the interface surface area. The macroscopic average of electrostatic potential $\overset{=}{V}$(z), which is employed as a reference level to align the valence-band, was calculated using the formula: V=(z)=1L∫−L2L2V¯(z)dz′´, where *L* is a corresponding distance of one period at each point $z^{\prime }.$ The planar-averaged potential $\bar{V}\left(z\right)$ and macroscopic average potentials $\overset{=}{V}$(z) of the ks-CZTS/CdS and st-CZTS/CdS heterostructures are shown in Appendix A, respectively. After determining the relative position of the valence bands, we simply add the calculated band gaps to obtain the discontinuities for the conduction $\Delta E_{c}=\Delta E_{v}+\Delta E_{g}$. The schematic energy band alignment of the ks-CZTS/CdS (Figure 4b) and st-CZTS/CdS (Appendix A) heterostructures reveal a staggered type II band alignment, in agreement with the experimental results (Figure 4a). Within the accuracy of our calculations, the CBO and VBO of the ks-CZTS/CdS heterostructure are predicted at 0.21 and 1.15 eV, respectively, whereas for the st-CZTS/CdS heterostructure, they are predicted at 0.28 and 1.24 eV (Appendix A), respectively. The staggered band alignment at the junction of ks-CZTS/CdS and st-CZTS/CdS heterojunctions with the VB_max_ and CB_min_ of CZTS higher than that of CdS suggest that photogenerated charge carriers will be efficient separated and transferred across the interface, where in both cases electrons will migrate to the CdS and holes to the CZTS. The predicted small CBO at the interfaces suggests lesser resistance for electron transport across the ks-CZTS/CdS and st-CZTS/CdS junctions. 

## 4. Conclusions

In summary, we have synthesized a single-phase kesterite-CZTS films with highly uniform surface (low roughness ~3.17 nm) and optimum band gap of 1.39 eV using RF sputtering followed by low-temperature H_2_S annealing. The band offsets of the CZTS/CdS heterojunction have also been systematically investigated by combining experiments with first-principles DFT calculations. It was demonstrated from CV measurements that a staggered type II band alignment with a small conduction band offset of ~0.1 eV (CV) exist at the CZTS/CdS interface, which is well validated by our first-principles DFT calculations. Atomic-level insights into the nature atomic ordering and stability of the CZTS/CdS interface reveal that the interface is composed mainly of strong covalent Cu–S and Cd–S interactions, resulting in a stable heterojunction. Our results point to efficient separation of photo-excited charge carriers across the CZTS/CdS interface and the predicted small CBO suggest that the CZTS/CdS heterojunction is ideal for fabricating efficient solar cells.

## Figures and Tables

**Figure 1 materials-12-04040-f001:**
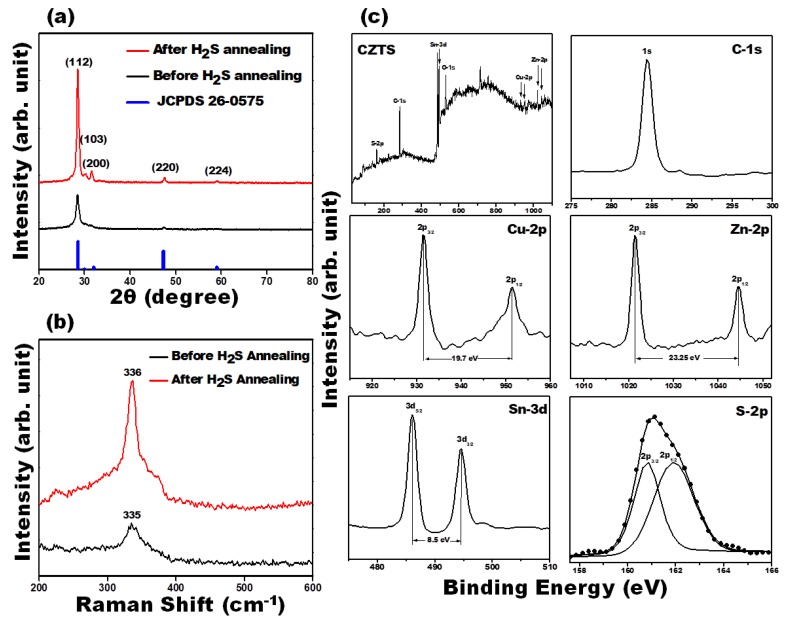
XRD pattern (**a**) Raman spectra (**b**) of as-deposited and H_2_S-annealed Cu_2_ZnSnS_4_ (CZTS) thin films and (**c**) High-resolution XPS spectrum of Cu_2_ZnSnS_4_ thin film. The corresponding peak positions of Cu_2_ZnSnS_4_ precursors show that these exhibit in Cu^+^, Zn^2+^, Sn^4+^, and S^−^ valence state.

**Figure 2 materials-12-04040-f002:**
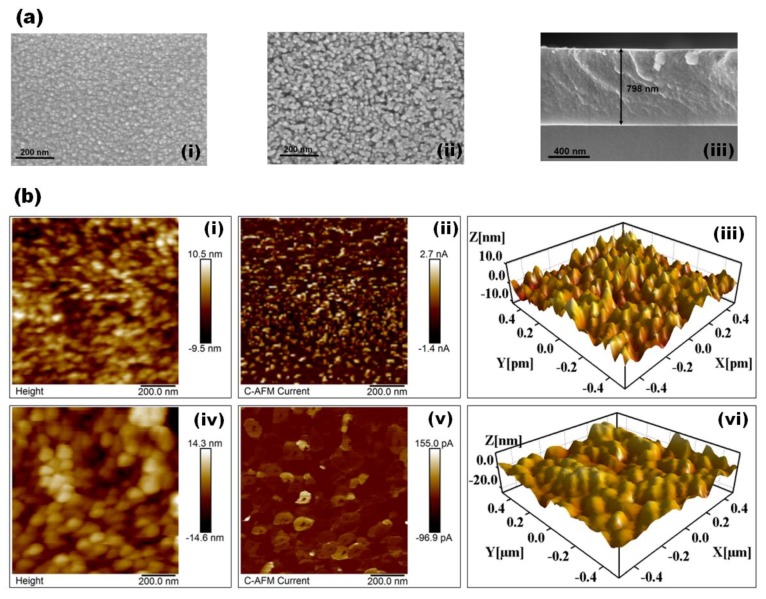
(**a**) (i) and (ii) FE-SEM images and (iii) cross-section FE-SEM images of as-deposited CZTS films. (**b**) (i) and (iv) surface topography, (ii) and (v) C-AFM images and (iii) and (vi) 3-D AFM images of as-deposited and H_2_S-annealed CZTS films, respectively.

**Figure 3 materials-12-04040-f003:**
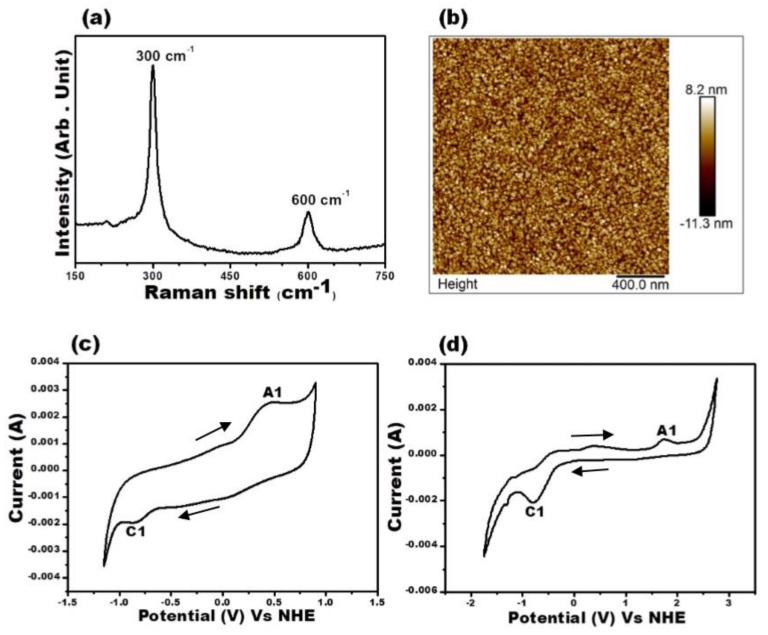
(**a**) Raman spectra of CdS thin film. (**b**) Atomic force microscopy (AFM) image for CdS thin film. Cyclic Voltammogram for CZTS after H_2_S annealing (**c**) and for CdS (**d**) recorded at scan rate of 100 mV/s. The arrows in the CV plot (**c**,**d**) indicate the direction of sweep from zero towards positive potential and then towards negative direction. The anodic and cathodic peaks are indicated by A_1_ and C_1_, respectively.

**Figure 4 materials-12-04040-f004:**
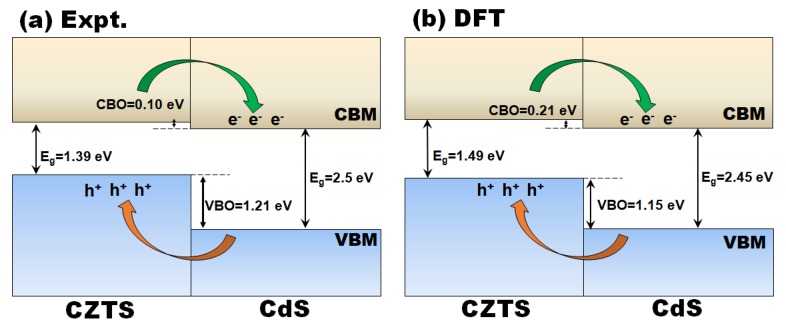
Energy band alignment diagram of the CdS/CZTS heterojunction from (**a**) experimental estimation and (**b**) DFT prediction.

**Figure 5 materials-12-04040-f005:**
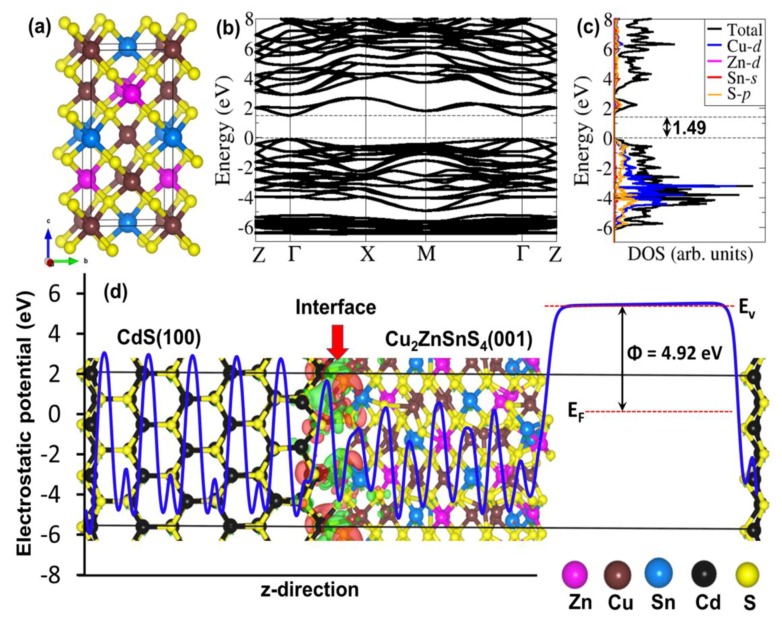
(**a**) Crystal structure, (**b**) band structure along the high-symmetry directions of the Brillouin zone, and (**c**) partial density of states (PDOS) of ks-CZTS. (**d**) Geometry optimized model of the CdS(100)/ks-CZTS (001) interface and the corresponding electrostatic potential (solid blue line), with the vacuum (E_v_) and Fermi (E_F_) level indicated by broken red line. Φ is the work function.

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
