# Peer review of "Interface Structure and Band Alignment of CZTS/CdS Heterojunction: An Experimental and First-Principles DFT Investigation"

_materials, 2019, doi:10.3390/ma12244040_

Round 1

Reviewer 1 Report

Manuscript is well written however the significance and novelty should be more strongly highlighted. In introduction the recent examples of multimetallic photocatalysts have to be discussed e.g. based on the following articles: 

1) International Journal of Hydrogen Energy, Volume 44, Issue 50,
2019, Pages 27343-27353, https://doi.org/10.1016/j.ijhydene.2019.08.173

2) Environ. Prog. Sustainable Energy 2019, 38: 13109. doi:10.1002/ep.13109

Authors ahould provide more detailed information about CV measurements in caption fig 3. e.g. pH of electrolyte, scan rate, direction of sweep.

Author Response

Responses Attached

Reviewer 2 Report

In this work the synthesis and characterization of a kesterite Cu2ZnSnS4 (CZTS) thin film has been presented. XRD, Raman and XPS measurements were performed to characterize and analyze the film.

Ab initio calculations were also employed to analyze the band offsets of the CZTS/CdS heterojunction. These corroborate and integrate the cyclic voltammetry (CV) experiments. Indeed, the conduction and valence bands offsets were estimated at 0.10 and 1.21 eV from CV measurements and these values are fairly reproduced by DFT calculations.

The paper is well designed and the exposition of the results is clear and convincing. It deserves publication in Materials.

Author Response

Fully in support of the publication of the manuscript in its current format. 
